# Organogenesis in a Broad Spectrum of Grape Genotypes and *Agrobacterium*-Mediated Transformation of the Podarok Magaracha Grapevine Cultivar

**DOI:** 10.3390/plants13192779

**Published:** 2024-10-03

**Authors:** Galina Maletich, Alexander Pushin, Evgeniy Rybalkin, Yuri Plugatar, Sergey Dolgov, Pavel Khvatkov

**Affiliations:** 1Federal State Funded Institution of Science “The Labor Red Banner Order Nikita Botanical Gardens–National Scientific Center of the RAS”, 298648 Yalta, Russia; 2Branch of Shemyakin and Ovchinnikov Institute of Bioorganic Chemistry, 142290 Puschino, Russia

**Keywords:** *Agrobacterium*-mediated transformation, benzyladenine, callus, grapevine, *Vitis*, elongation, morphogenesis, regeneration

## Abstract

We present data on the ability for organogenesis in 22 genotypes of grapevine and developed a direct organogenesis protocol for the cultivar Podarok Magaracha and the rootstock Kober 5BB. The protocol does not require replacement of culture media and growth regulators, and the duration is 11 weeks. The cultivation of explants occurs on modified MS medium with the addition of 2.0 mg L^−1^ benzyladenine and indole-3-butyric acid (0.15 mg L^−1^ for the rootstock Kober 5BB or 0.05 mg L^−1^ for the cultivar Podarok Magaracha). The direct organogenesis protocol consists of three time periods: (1) culturing explants for 2 weeks in dark conditions for meristematic bulk tissue, (2) followed by 4 weeks of cultivation in light conditions for regeneration, and (3) 5 weeks of cultivation in dark conditions for shoot elongation. Based on this protocol, conditions for the *Agrobacterium*-mediated transformation of the Podarok Magaracha cultivar were developed with an efficiency of 2.0% transgenic plants per 100 explants. Two stably transformed lines with integration into the genome of the *pBin35SGFP* plasmid construction, confirmed by Southern blotting, were obtained.

## 1. Introduction

Grapevine (*Vitis* spp.) is one of the most common and economically important fruit crops in the world. The use of genetic engineering to improve grapevine makes it possible to introduce useful agrotechnical traits without changing the properties of the cultivar. The practical application of conventional breeding methods for grapevines has proved to be problematic, mainly because of their long juvenile period and reproductive cycle and high level of heterozygosity and the introgression of undesirable genes during crossbreeding, which can affect the agronomic performance and quality of a cultivar. Genetic transformation could help to overcome these drawbacks [1]. The effectiveness of tissue culture techniques for the genetic transformation of grapevine depends on the availability of highly reproducible and efficient in vitro regeneration protocols [2]. Regenerative ability is a feature unique not only to plant species or cultivars but also to individual plants, and, as such, it is a characteristic of a plant’s genotype [3,4,5,6,7,8,9].

The ability of plants to regenerate is affected by multiple factors, including the use of plant growth regulators [10,11], the composition of the basal medium, including the salt and vitamin contents [12,13], and the explant type [14,15]. Currently, the regeneration and transformation of a grapevine can be achieved using different types of explants, both through shoot organogenesis and through somatic embryogenesis [16].

Somatic embryogenesis and regeneration in whole plants were first described by Mullins and Srinivasan [17]. Regeneration in grapevine in vitro is mainly achieved via somatic embryogenesis [18,19,20,21,22,23,24]. However, achieving regeneration and transformation through embryogenic cultures is difficult and is restricted to a few genotypes. The process demands the continuous induction and maintenance of embryonic cultures, requiring intensive labor, time, and space, as well as patience and skill [25]. Grapevine plants can also be regenerated via organogenesis [26,27]. For this purpose, fragments of shoot tips [28], fragments of internodes [29,30,31,32], fragments of leaves [32,33,34,35,36,37,38,39], and fragments of petioles [32,40,41] are often used as explants. However, reports on the use of direct organogenesis in genetic transformation from somatic tissues are extremely rare [1,25]. In 2002, Mezzetti et al. [42] reported on the genetic transformation of *Vitis vinifera* via organogenesis through meristematic bulk induction produced in the presence of naphthaleneacetic acid (NAA) and benzyladenine (BA), and they used this technique to genetically engineer the table grapevine cultivars Silcora and Thompson Seedless.

The aim of our research was to develop a protocol for plant regeneration in 22 grapevine genotypes, including 11 vine grape cultivars of significance in the global wine industry (Merlot, Pinot Noir, Pinot Gris, Ruta, Sphinx, Cabernet Sauvignon, Aligote, Syrah, Bastardo, Muscat Blanc, and Chardonnay), 1 important cultivar in the global wine industry rootstock (Kober 5BB), and 10 local cultivars (Podarok Magaracha, Kefesiya Magaracha, Akademik Avidzba, Yaltinskiy bessemyannyy, Magarach no. TT2, Muscat Crima, Veles, Tsitronnyy Magaracha, Livia, and Krymskiy biser), for which there is generally a lack of experimentation in the scientific literature regarding their use in genetic engineering via *Agrobacterium tumefaciens*.

## 2. Results

### 2.1. Development of a Regeneration Protocol

Over the course of the experiments, it was found that, apparently due to the high genetic diversity of different grapevine cultivars, the conditions for the induction of direct organogenesis among them are significantly different.

As a result of studying the influence of the growth regulators thidiazuron (TDZ) and indole-3-butyric acid (IBA) on the efficiency of grapevine regeneration, results showed low efficiency for 1 breeding form (Magarach no. TT2—1.7%) and 13 cultivars (Livia—3.5 ± 1.8%, Veles—1.7 ± 0.4%, Tsitronnyy Magaracha—0.0%, Muscat Blanc—0.0%, Sphinx—0.0%, Pinot Noir—1.7 ± 0.3%, Merlot—1.7 ± 0.3%, Syrah—3.5 ± 1.8%, Chardonnay—0.0%, Cabernet Sauvignon—1.7 ± 0.3%, Krymskiy biser—3.5 ± 1.8%, Pinot Gris—0.0%, and Bastardo—3.5 ± 1.8%). Figure 1 shows the regeneration efficiency of the studied grape genotypes above 10%; five cultivars (Yaltinskiy bessemyannyy—19.3 ± 2.5%, Aligote—10.0 ± 1.4%, Akademik Avidzba—9.8 ±3.6%, Kefesiya Magaracha—9.8 ± 4.2%, and Muscat Crima—10.0 ± 1.5%), and one rootstock (Kober 5BB—25.5 ± 3.0%) were regenerated with an efficiency of 9.8–25.5%. Two cultivars showed high regeneration potential (Ruta—87.0 ± 6.2% and Podarok Magaracha—100.0 ± 7.2%).

In the genotypes Yaltinskiy bessemyannyy and Magarach no. TT2, the greatest regeneration was observed when using leaf fragments cultivated on MS medium with the addition of 0.5 mg L^−1^ TDZ together with 0.1 mg L^−1^ IBA as explants. When the TDZ concentration increased to 1.0 or 1.5 mg L^−1^, the regeneration efficiency of the Yaltinskiy bessemyannyy cultivar decreased from 19.3 to 3.7 or 0.0%, respectively, and, for the Magarachsky TT2 breeding form, it decreased to 0.0%. When the IBA concentration was reduced to 0.05 or 0.01 mg L^−1^, no regeneration of either genotype was observed. In the Akademik Avidzba cultivar, the greatest regeneration efficiency was observed when using explants of the leaf fragments, and, in the Aligote cultivar, this was achieved when using the explants of internode fragments, cultured on MS medium with a combination of the growth regulators 1.0 mg L^−1^ TDZ and 0.1 mg L^−1^ IBA. Increasing or lowering the TDZ concentration or lowering the IBA concentration below 0.1 mg L^−1^, resulted in a regeneration efficiency of 0.0% in both cultivars. Regeneration of the Muscat Crima cultivar and Kober 5BB rootstock was most effective when petiole fragments were used as explants, when the growth regulator 0.01 mg L^−1^ IBA was added to the medium, together with 0.5 and with 1.5 mg L^−1^ TDZ, respectively. An increase in IBA concentration to 0.05 and 0.1 mg L^−1^ led to a decrease in the efficiency of regeneration in the Muscat Crima cultivar from 10.0 to 0%, and the Kober 5BB rootstock saw a decrease in regeneration efficiency from 25.5 to 2.4% at 0.05 mg L^−1^ IBA, and to 6.7% at 0.1 mg L^−1^ IBA. An increase in TDZ concentration to 1.0 mg L^−1^ in the medium led to the absence of regeneration in the Muscat Crima cultivar. A decrease in the TDZ concentration to 1.0 mg L^−1^ caused a decrease in the regeneration efficiency of the Kober 5BB rootstock by 3.0%, and a decrease in the TDZ concentration to 0.5 mg L^−1^ led to a decrease in regeneration efficiency of 24.3%. The greatest regeneration efficiency of the Kefesia Magaracha cultivar was achieved using leaf fragment explants (9.8%) cultivated on MS medium with the addition of 1.5 mg L^−1^ TDZ, together with 0.1 mg L^−1^ IBA. A high regenerative ability was shown by the cultivar Ruta using a petiole fragment as the explant (87.0%) and by Podarok Magaracha when using explants of leaf fragments (up to 100%). The optimal combination of growth regulators for the regeneration of the Ruta cultivar is 1.0 mg L^−1^ TDZ, together with 0.05 mg L^−1^ IBA. Reducing the TDZ concentration to 0.5 mg L^−1^ reduces the regeneration efficiency by 40.4%, and increasing the TDZ concentration to 1.5 mg L^−1^ reduces the regeneration efficiency by 26.8%. A similar effect is caused by an increase or decrease in the concentration of IBA in the medium. When the IBA content is 0.01 mg L^−1^ in the medium, the regeneration efficiency decreases from 87.0 to 28.4%, and, at a concentration of 0.1 mg L^−1^, it decreases from 87.0 to 50.0%. In the Podarok Magaracha cultivar, the highest regeneration was observed using the growth regulator TDZ, from 1.0 to 1.5 mg L^−1^, together with IBA from 0.05 to 0.1 mg L^−1^: the regeneration efficiency reached 100%. There were no statistically significant differences (Appendix A).

As a result of studying the influence of the growth regulators BA and IBA on the efficiency of grapevine regeneration, the following results were found for 11 grapevine cultivars (Livia—0.0%, Veles—0.0%, Tsitronnyy Magaracha—1.1 ± 0.3%, Muscat Blanc—0.0%, Sphinx—0.0%, Pinot Noir—1.7 ± 0.3%, Aligote—1.1 ± 0.2%, Merlot—0.0%, Chardonnay—1.1 ± 0.3%, Cabernet Sauvignon—1.1 ± 0.2%, and Pinot Gris—0.0%) and 1 breeding form (Magarach no. TT2—1.1 ± 0.4%). Figure 2 shows the regeneration efficiency of the studied grape genotypes that exceeded 10%. For four cultivars (Kefesiya Magaracha—13.0 ± 1.2%, Muscat Crima—12.6 ± 1.5%, Ruta—13.3 ± 2.5%, and Syrah—15.3 ± 1.1%), the regeneration efficiency ranged from 12.0 to 15.0%. In four cultivars (Bastardo—33.4 ± 3.4%, Yaltinskiy bessemyannyy—50.0 ± 6.1%, Akademik Avidzba—43.2 ± 5.3%, and Krymskiy biser—53.4 ± 6.2%) and a rootstock (Kober 5BB—53.3 ± 6.2%), the regeneration efficiency ranged from 33.0 to 53.5%. The Podarok Magaracha cultivar had the highest regeneration efficiency of 87.4 ± 7.8%.

Leaf fragments of the cultivars Syrah, Muscat Crima, and Akademik Avidzba showed the greatest regeneration potential cultivated with the adaxial side on initiation medium (IM) with the addition of 3.0 mg L^−1^ BA, together with 0.15 mg L^−1^ IBA. A decrease in concentration from 3.0 to 2.5 mg/l of BA led to a decrease in regeneration efficiency in the Syrah cultivar from 15.3 to 7.8%, in the Muscat Crima cultivar from 12.6 to 0.0%, and in the Akademik Avidzba cultivar from 43. 2 to 36.5%. Reducing IBA to 0.1 mg L^−1^ had a similar effect on regeneration efficiency. The regeneration efficiency of the Syrah cultivar was 2.7%; for the Muscat Crima cultivar, it was 0.0%, and for Akademik Avidzba, it was 20.0%. These results are, respectively, 12.6, 12.6, and 23.2%, lower than those seen using 0.15 mg L^−1^ IBA. In the cultivars Kefesia Magaracha and Krymskiy biser, the greatest regeneration efficiency was observed using leaf fragments as explants cultivated on IM with the growth regulators 2.5 mg L^−1^ BA and 0.15 mg L^−1^ IBA. Both an increase and a decrease in the concentration of BA and/or IBA in the IM led to a decrease in the efficiency of regeneration in the Yaltinskiy bessemyannyy cultivar, which was most effective with the use of leaf fragments as explants (50.0%) cultivated on IM with the addition of 3.0 mg L^−1^ BA and 0.1 mg L^−1^ IBA. A decrease in BA in the medium to 2.5 mg L^−1^ reduced the regeneration efficiency by 46.5%. Reducing the IBA content in the medium to 0.05 mg L^−1^ reduced the regeneration efficiency by 37.5%. Regeneration of the Ruta cultivar was most effective when petiole fragments (13.3%) were used as explants, cultivated on a medium supplemented with the growth regulator 2.5 mg L^−1^ BA, together with 0.05 mg L^−1^ IBA. A decrease or increase in the BA content in the medium to 2.0 or 3.0 mg L^−1^ led to a lack of regeneration. Increasing the concentration of IBA in the medium reduced the regeneration efficiency from 13.3 to 6.3%. The greatest efficiency of regeneration seen in the Kober 5BB rootstock was when petiole fragments were used as explants (53.3%), and the greatest regeneration efficiency was achieved in the Podarok Magaracha cultivar when leaf fragments were used as explants (87.4%). The best combination of growth regulators for the regeneration of Kober 5BB rootstock is 2.0 mg L^−1^ BA, together with 0.15 mg L^−1^ IBA. Increasing the BA concentration to 2.5 mg L^−1^ reduced the regeneration efficiency from 53.3 to 25.4%. Reducing the IBA concentration in the medium to 0.1 mg L^−1^ also led to a decrease in regeneration from 53.3 to 23.8%. For the regeneration of the Podarok Magaracha cultivar, the best combination of growth regulators is 2.0 mg L^−1^ BA, together with 0.05 mg L^−1^ IBA. An increase in the BA concentration in the medium to 2.5 mg L^−1^ reduced the regeneration efficiency by 10.6%, and an increase in the IBA concentration to 0.1 mg L^−1^ caused a decrease in the regeneration efficiency of 24.1% (Appendix A).

Logistic regression models were built only for genotypes that showed regeneration (shown in Figure 1 and Figure 2). It was not possible to train a model for the following genotypes from Appendix A: Cabernet Sauvignon, Chardonnay, Merlot, Pinot Noir, Pinot Gris, Bastardo, Muscat Blanc, Citroniy Magaracha, Veles, Magarach No. TT2, and Sphinx; and from Appendix A: Cabernet Sauvignon, Chardonnay, Merlot, Pinot Noir, Pinot Gris, Aligote, Muscat Blanc, Citroniy Magaracha, Veles, Magarach No. TT2, Liviya, and Sphinx. In Appendix A, *p* values greater than 0.05 are highlighted in bold, indicating that the corresponding factors do not exert a statistically significant effect on the result and there is no reason to believe that this factor is important in predicting the result (the probability of regeneration for the model of the corresponding genotype). Most often, such a result corresponded to the “auxin concentration” factor, for models of 7 genotypes in Appendix A and 9 genotypes in Appendix A. In second place is the “cytokinin concentration” factor with models of 4 and 1 genotypes, and in third place is the “explant type” factor with models of 2 and 1 genotypes from Appendix A, respectively. The confidence interval (95%) does not cross 0 for any model, indicating a significant influence of the factors (provided that *p* < 0.05), and that this influence is not random.

In general, all cultivars can be clustered according to the effectiveness of their regenerative ability. The first cluster consists of 14 genotypes (Livia, Veles, Tsitronnyy Magaracha, Muscat Blanc, Sphinx, Aligote, Pinot Noir, Merlot, Chardonnay, Cabernet Sauvignon, Pinot Gris, Kefesiya Magaracha, Muscat Crima, and Magarach no. TT2). The regeneration efficiency of these genotypes is at a low level (no more than 13.0%), regardless of the combination of growth regulators used. The second cluster consists of five cultivars (Syrah, Bastardo, Akademik Avidzba, Krymskiy biser, and Yaltinskiy bessemyannyy). These cultivars regenerate at a medium level with the combination of BA and IBA (with the regeneration efficiency ranging from 15.0 to 53.0%) and at a low level with the combination of the growth regulators TDZ and IBA (with the regeneration efficiency ranging from 3.5 to 10.0%). Rootstock Kober 5BB has a high regenerative ability in the presence of the growth regulators BA and IBA (53.3 ± 6.2%); in the presence of TDZ and IBA, this regeneration efficiency decreases to 25.5 ± 3.0% (see Appendix A; the data are valid with petiole fragments as explants). The Podarok Magaracha cultivar is characterized by a high regenerative ability, both with a combination of the growth regulators BA and IBA and with TDZ and IBA; the regeneration efficiency is 87.4 ± 7.8 and 100.0 ± 7.2%, respectively (see Appendix A; the data are valid with leaf fragments as explants). The Ruta cultivar has a high regenerative ability using a combination of the growth regulators TDZ and IBA (87 ± 6.2%) and a low regenerative ability in the presence of BA and IBA (13.3 ± 2.5%) (see Appendix A; the data are valid with petiole fragments as explants).

In our experiments, the regenerants obtained on the MS medium with the addition of TDZ and IBA mostly showed deformation and vitrification of the shoot, whereas when using IM with the addition of BA, and IBA, normal shoot formation was observed (Figure 3).

### 2.2. Enhancing Efficiencies for the Most Responsible Genotypes

Our further research focused on increasing the regeneration efficiency of two grapevine genotypes (the rootstock Kober 5BB and the cultivar Podarok Magaracha), which showed the greatest regenerative ability when using a combination of the growth regulators BA and IBA. To achieve this, we studied the influence of various culture media and the influence of the duration of pre-cultivation under dark conditions on the induction of grapevine regeneration.

In testing various culture media (IM, PIV, NN, WPM, DKW, and MS) to induce grapevine shoot regeneration, it was noted that the greatest regeneration efficiency in both grapevine genotypes was observed using the IM (Table 1). The regeneration efficiency of the Kober 5BB rootstock was 52.1 ± 6.4% (IM, supplemented with 2.0 mg L^−1^ BA and 0.15 mg L^−1^ IBA), and, for the Podarok Magaracha cultivar, this was 97.6 ± 7.5% (IM, supplemented with 2.0 mg L^−1^ BA and 0.05 mg L^−1^ IBA).

When studying the duration of pre-cultivation of explants under dark conditions, it was found that regeneration from explants of both grapevine genotypes (the rootstock Kober 5BB and the cultivar Podarok Magaracha) occurred in the third week of cultivation (Figure 4a,b). The regeneration of explants of the Kober 5BB rootstock without pre-cultivation of the explants under dark conditions began on the 19th day and ended on the 24th day of cultivation (the duration of the regeneration period was 5 days), and the regeneration efficiency was 28.6 ± 5.4% (Figure 4a). The regeneration from explants pre-cultured for one week under dark conditions began on the 10th day after transferring the explants to light conditions (the 17th day of cultivation) and ended on the 19th day (the 26th day of cultivation) (Figure 4a). The duration of the regeneration period was 9 days, and the regeneration efficiency was 31.0 ± 3.0% (Figure 4a). The regeneration of shoots from explants of other experimental variants (pre-cultivation for 2, 3, 4, and 5 weeks) began on the third day after transferring the explants from dark conditions to light (17, 24, 31, and 38 days of cultivation, respectively). Explants of the Kober 5BB rootstock pre-cultivated for 2 weeks showed the highest regeneration efficiency (53.3 ± 5.4%); the duration of the regeneration period was 19 days (Figure 4a). For the variants of the pre-cultivation of explants of 3, 4, and 5 weeks, the regeneration efficiency was 42.0 ± 3.8, 13.1 ± 2.6, and 10.0 ± 1.9%, respectively (Figure 4a), and the duration of the regeneration period was reduced to 13, 6, and 6 days, respectively (Figure 4a).

The regeneration of explants of the Podarok Magaracha cultivar without the pre-cultivation of the explants under dark conditions began on the 21st day and ended on the 35th day of cultivation (the duration of the regeneration period was 14 days). The regeneration efficiency was 19.4 ± 2.0% (Figure 4b). The regeneration of explants pre-cultured for one week under dark conditions began on the 9th day after transferring the explants to light conditions (the 16th day of cultivation) and ended on the 14th day (the 21st day of cultivation) (Figure 4b). The duration of the regeneration period was 5 days, and the regeneration efficiency was 94.9 ± 5.0% (Figure 4b). Regeneration from explants pre-cultured for 2 weeks under dark conditions began on the 5th day after transferring the explants to light conditions (the 19th day of cultivation) and ended on the 7th day (the 21st day of cultivation). The duration of the regeneration period was 2 days, and the regeneration efficiency was 95.5 ± 5.3%. The regeneration of other explants (pre-cultured for 3, 4, and 5 weeks) occurred on the second day after transferring the explants from dark conditions to light (23, 30, and 37 days of cultivation, respectively) (Figure 4b). The duration of pre-cultivation of explants under dark conditions did not affect the efficiency of regeneration. The regeneration efficiency of all explants pre-cultured in the dark ranged from 93.3 ± 4.5 to 97.6 ± 6.4% (Figure 4b). The regeneration period lasted from 2 to 4 days from the moment that the leaf explants were transferred to light cultivation conditions.

### 2.3. Development of an Elongation Protocol

The resulting shoots could not be separated from the explant tissues, due to their small size. To elongate shoots, explants were placed in the dark. Elongated etiolated shoots developed at 5 weeks of cultivation in the dark (Figure 5). The elongation efficiency ranged from 5.6 to 6.2% for the Kober 5BB rootstock and from 15.8 to 16.1% for the Podarok Magaracha cultivar, with a regeneration efficiency of 55.0–58.4% and 91.6–93%, respectively. There was no statistically significant effect of the duration of cultivation under light conditions (3 or 4 weeks of cultivation) on the efficiency of shoot elongation.

### 2.4. Tissue-Culture Cycle (Direct Organogenesis)

Figure 6 summarizes the protocol for direct organogenesis from somatic tissues of grapevine [rootstock Kober 5BB (Figure 6) and cultivar Podarok Magaracha (Figure 6)]. This protocol does not require the replacement of culture media and growth regulators (IM 3% (*w*/*v*) sucrose, 0.75% (*w*/*v*) agar, with the addition of the growth regulators 2.0 mg L^−1^ BA and 0.15 mg L^−1^ IBA for rootstock Kober 5BB or 2.0 mg L^−1^ BA and 0.05 mg L^−1^ for the cultivar Podarok Magaracha); the duration is 11 weeks. The direct organogenesis protocol consists of culturing explants for 2 weeks under dark conditions to obtain the meristematic bulk tissue, followed by 4 weeks of cultivation under light conditions for regeneration and 5 weeks of cultivation under dark conditions for shoot elongation.

### 2.5. Agrobacterium-Mediated Transformation of ‘Podarok Magaracha’

After the inoculation of pre-cultured leaf explants with *Agrobacterium* harboring the *pBin35SGFP* construct, the transient expression of the green fluorescent protein (GFP) gene was detected after 48 h. The peak level of transient expression was observed on days 4–6 after transformation (highlighted in red, in Table 2). Then, the number of explants with transient expression decreased. After 70 days, it was established that after the inoculation of explants with a suspension of *Agrobacterium* (optical density (OD)600 = 1.0) and subsequent co-cultivation for 72 and 96 h, the efficiency of GFP expression reached a maximum of 27.6 and 29.3%, respectively (according to Duncan’s test, these data belong to cluster “r” (Table 2)). The process of the formation of calli expressing GFP was accompanied by the necrotization of explant tissue. To reduce necrotization and transformation stress in explant tissues, we recommend using an agrobacterial suspension with a lower concentration (OD_600_ = 0.8) and co-cultivating explants with *Agrobacterium* for 72 h, since with longer co-cultivation (96 h), no statistically significant increase in transformation efficiency was found (according to Duncan’s test, these data belong to cluster “pq” (Table 2)). In this experiment, no transgenic plants were obtained. It is obvious that the combinations of growth regulators used for the regeneration of non-transgenic tissues are not optimal for the formation of transgenic shoots from competent callus cells.

Based on the developed direct organogenesis protocol, we optimized the concentrations and combinations of growth regulators added to the medium for the regeneration of transgenic ‘Podarok Magaracha’ shoots (Table 3).

As a result of transformation, nine shoots expressing GFP were obtained (Figure 7a), two of which were able to elongate under dark cultivation conditions (Figure 7c), and then cut from the explant and rooted on plant growth (PG) medium supplemented with 0.05 mg L^−1^ 1-naphthalene acetic acid (NAA), 150 mg L^−1^ timentin (Tm), and 50 mg L^−1^ kanamycin sulfate (Km), to obtain whole plants (Figure 7f,g). The most effective combination of growth regulators for the regeneration of transgenic shoots was 2.5 mg L^−1^ BA and 0.4 mg L^−1^ IAA (Table 3).

According to the polymerase chain reaction (PCR) analysis, both transgenic lines contained the target *gfp* and neomycin phosphotransferase (*npt*ll) heterologous DNA (Figure 8). Contamination by the *Agrobacterium vir* gene was not detected in the analyzed samples. To further confirm the transgenic origin of these lines, Southern blot analysis was performed. This result confirmed the integration of *gfp* into the grapevine genome. Between one and two transgene copies were revealed in the analyzed lines; for example, line PM_GFP1/2 had one copy, and line PM_GFP1/1 had two copies (Figure 9). As expected, no hybridization signal was observed in the analysis of wild-type grapevine plants, and transgenic lines revealed *gfp* expression (Figure 7f,g). As a result, we obtained 2 whole transgenic plants from 100 inoculated explants of the cultivar Podarok Magaracha.

## 3. Discussion

The aim of our research was to develop a protocol for plant regeneration in 22 grapevine genotypes, regarding its use in genetic engineering via *Agrobacterium tumefaciens*. It is a well-known fact that the regeneration of shoots from an explant in vitro is influenced by many factors, such as the composition of the culture medium, growth regulators, the plant genotype, the type of explant used, and other cultivation conditions. Our study examined the effect of two cytokinins, TDZ and BA, together with IBA, in various combinations, on the regeneration efficiency of 22 grapevine genotypes. Of these, three grapevine genotypes (Ruta, Podarok Magaracha, and Kober 5BB) showed a high regeneration efficiency. The inducing role of both TDZ and BA in grape regeneration has been noted by a broad range of researchers. In research by Zhang et al. [32], callus induction and plant regeneration from leaf explants of the Wink cultivar of grapevine were achieved using 4.0 mg L^−1^ TDZ and 0.1 mg L^−1^ IBA (with a regeneration efficiency of 78.7%). Most researchers report the effectiveness of BA for inducing grape regeneration. Kumsa and Feyissa [43] reported regeneration from leaf explants of the grape cultivars Chenin Blanc and Canonannon using combinations of the growth regulators 2.0 mg L^−1^ BA and 0.1 mg L^−1^ IBA (the regeneration efficiency was 86.0% and 88.4%, respectively). Li et al. [41] showed callus induction and regeneration in the Kober 5BB rootstock from petiole explants using the growth regulators 2.5 mg L^−1^ BA and 0.05 mg L^−1^ IBA, with a regeneration efficiency of 43.3%; these data are similar to our results (Appendix A). A study by Clog et al. [33] reported the regeneration of Kober 5BB rootstock from leaf explants; the regeneration efficiency was 38.6%. Apparently, the type of explant plays an important role in the regeneration of grapevine shoots, and for the Kober 5BB rootstock, it is preferable to use petiole fragments as an explant. We suggest that the combination of the growth regulators BA and IBA is suitable for direct organogenesis in grapevines. However, the use of a combination of the growth regulators BA and IBA, even when using different types of explants, cannot be considered universal for grapevine regeneration. Among the 22 cultivars we studied, 10 did not show morphogenic abilities (Merlot, Pinot Noir, Pinot Gris, Sphinx, Cabernet Sauvignon, Muscat Blanc, Chardonnay, Veles, Tsitronnyy Magaracha, and Livia). In total, four cultivars (Syrah, Kefesiya Magaracha, Muscat Crima, and Aligote) and one breeding form (Magarach no. TT2) showed a regeneration efficiency below 15.0%. The regeneration efficiency of the Bastardo, Akademik Avidzba, Yaltinskiy bessemyannyy, and Krymskiy biser cultivars ranged from 33.0 to 53.0%. This may be due to differences in the genotypes used, as different plant growth regulators at different concentrations show significant variability in in vitro regeneration among different species or cultivars. Plant growth regulators that are effective for one cultivar may not be as effective for another cultivar [44,45].

The type of explant used has a significant impact on the efficiency of regeneration. [26]. In the present study, we studied the regeneration of leaf explants, petioles, and internodes. We found that for most study genotypes (Syrah, Podarok Magaracha, Kefesiya Magaracha, Akademik Avidzba, Yaltinskiy bessemyannyy, Magarach no. TT2, Muscat Crima, and Krymskiy biser), the leaf is preferable as an explant to the petiole and internode (Appendix A). This is consistent with research by Quan and Chang [46] and Zhang et al. [32]. However, three genotypes (Ruta, Bastardo, and Kober 5BB) showed the highest regenerative abilities when using petiole fragments as explants, and the Aligote cultivar showed the highest regenerative abilities when using internode fragments.

The significance of the influence of the mineral composition of the culture medium on shoot regeneration in various grapevine cultivars is a controversial issue. For example, Kurmi et al. [31] believe that the composition of the culture medium does not play as important a role in the in vitro explant response as the type and concentration of plant growth regulators. At the same time, a study by Mezzetti et al. [42] demonstrated the successful use of a modified MS medium for initiation and shoot regeneration from grapevine somatic tissues. Our study shows that the mineral composition of the culture medium has a significant impact on the efficiency of regeneration (Table 2).

In our study, we examined the effect of the duration of explant cultivation under dark conditions on the regeneration efficiency of two grape genotypes and found that two weeks is sufficient to achieve an optimal regeneration efficiency (Figure 4), which is consistent with reports by Korban et al. [47], Toreegrosa et al. [48], Deng et al. [49], and Zhang et al. [32]. In a study by Kumsa and Feyissa [43], a four-week deposition of explants in the dark was necessary for shoot regeneration. In our study, cultivating explants for 4 weeks in the dark reduced the efficiency of regeneration in the Kober 5BB rootstock by 40%, but did not affect that of the Podarok Magaracha cultivar (Figure 4). This is probably also due to the properties of the genotype.

In our study, we were faced with the problem of shoot elongation in the obtained regenerated plants, which was solved in a simple way, without introducing new growth regulators into the cultivation medium. However, the elongation efficiency was 6.2% for the Kober 5BB rootstock and 16.1% for the Podarok Magaracha cultivar. In studies by Zhang et al. [32], Kumsa and Feyissa [43], Mezzetti et al. [42], and Xie et al. [25], no such problems were reported. This is likely due to differences in the genotypes used.

Transgenic grape plants are typically regenerated from somatic embryos derived from either zygotic embryos or leaves [50,51], or, more recently, from anther filaments [21]. In our method, transgenic grape plants were regenerated via organogenesis. For the success of *Agrobacterium*-mediated transformation, the competence of transformed cells is a critical aspect, producing tissue composed of cells having a high regenerative capacity and a large number of damaged cells. In our study, the protocol for regeneration and transformation through organogenesis was first applied to the local wine grape cultivar Podarok Magaracha.

From the results obtained, it seems that the transformation efficiency is not linked to the initial conditions for plant regeneration, which is consistent with the results of a study by Sabbadini et al. [1]. Sabbadini et al. [1] showed that the use of growth regulators necessary for plant regeneration contributed to the regeneration of transgenic shoots only in the Thomson Seedles cultivar; other genotypes formed only callus tissues after transformation (similar to the Podarok Magaracha cultivar in our study (Table 2)). We assume that the regeneration of transgenic shoots of some grape genotypes requires other (i.e., not the same as that for regular plant regeneration) combinations of growth regulators (Table 3). A separate study on the conditions necessary for the regeneration of transgenic cells may represent a solution to the problem of modifying different grape cultivars of significance in the global wine industry.

## 4. Materials and Methods

### 4.1. Plant Materials

To curate a research collection of grapevines in vitro, vines of twenty cultivars (Merlot, Pinot Noir, Pinot Gris, Ruta, Sphinx, Cabernet Sauvignon, Aligote, Syrah, Bastardo, Muscat Blanc, Chardonnay, Podarok Magaracha, Kefesiya Magaracha, Akademik Avidzba, Yaltinskiy bessemyannyy, Muscat Crima, Veles, Tsitronnyy Magaracha, Livia, and Krymskiy biser), one rootstock (Kober 5BB), and one breeding form (Magarach no. TT2) were collected from field-grown mother vines at the All-Russian National Scientific Research Institute of Vine And Winemaking “Magarach” (lat.: 44.850984′ N, long.: 33.650112′ E), vernalized for 1 month at a temperature of +4 °C, and germinated in vessels with water. Further, in the greenhouse, growing green shoots were superficially sterilized by submersion in sodium hypochlorite solution (2.5% NaOCl, 0.1% Tween 20) for 15 min and then washed fourfold with sterile water. The shoots were then cut into single-node cuttings and placed in tubes on modified MS, according to Zlenko et al. [24] (described as “plant growth” medium (PG); this contained KNO_3_ (922.0 mg L^−1^), NH_4_NO_3_ (308.8 mg L^−1^), KH_2_PO_4_ (82.0 mg L^−1^), MgSO_4_ × 7 H_2_O (597.0 mg L^−1^), and CaCl_2_ (331.0 mg L^−1^), vitamins from MS, Fe ethylenediaminetetraacetic acid (FeEDTA), and microelements halved via MS), supplemented with 1% (*w*/*v*) sucrose and 0.75% (*w*/*v*) agar (Panreac, Spain) and cultivated at pH 5.7, 25 ± 1 ° C and a light intensity of 65 µmol m^−2^ s^−1^ during a 16 h day photoperiod. After 2 months’ cultivation on PG medium from single-node cuttings (without signs of bacterial or fungal infection), shoots developed, which were cut off and placed in culture vessels (total volume: 500 mL) containing 50 mL agarized PG medium supplemented with 0.05 mg L^−1^ NAA, with seven plants per culture vessel.

### 4.2. Development of a Regeneration Protocol

The first two experiments on the regeneration of 22 grapevine genotypes were performed on Murashige and Skoog (MS) [52] and modified MS media, according to Mezzetti et al. [42] (described as “initiation medium” (IM); this contained KNO_3_ (1050 mg L^−1^), NH_4_NO_3_ (400 mg L^−1^), KH_2_PO_4_ (200 mg L^−1^), MgSO_4_ × 7 H_2_O (400 mg L^−1^), CaNO_3_ (750 mg L^−1^), NaH_2_PO_4_ (200 mg L^−1^), microelements, and vitamins from MS), supplemented with 0.75% (*w*/*v*) agar (Panreac, Spain) and various concentrations and combinations of the growth regulators BA and TDZ, jointly with IBA (Figure 10).

Experiment no. 1 used MS medium supplemented with 2% (*w*/*v*) sucrose and TDZ ranging from 0.5 to 1.5 mg L^−1^ in increments of 0.5 mg L^−1^, with the addition of IBA at 0.01, 0.05, or 0.1 mg L^−1^ [32]. To induce regeneration, the explants were pre-cultivated under dark conditions (3 weeks for leaf fragments and 2 weeks for petiole and internode fragments) at 24 ± 0.5 °C.

Experiment no. 2 used IM supplemented with 3% (*w*/*v*) sucrose and BA ranging from 2.0 to 3.0 mg L^−1^ in increments of 0.5 mg L^−1^, with IBA ranging from 0.05 to 0.15 mg L^−1^ in increments of 0.05 mg L^−1^ [39]. To induce regeneration, explants were pre-cultivated under dark conditions (4 weeks for leaf fragments and 3 weeks for petiole and internode fragments) at 24 ± 0.5 °C.

Fragments of leaves, petioles, and internodes obtained from aseptic in vitro grapevine plants cultivated on PG medium supplemented with 0.05 mg L^−1^ NAA were used as explants. The leaves were cut into fragments measuring 5 mm × 5 mm and placed in Petri dishes on the adaxial side. Petioles and internodes were cut into fragments 5 mm long and placed longitudinally on the culture medium. All explants were placed in Petri dishes containing 25 mL culture medium supplemented with different combinations of growth regulators (according to experimental conditions no. 1 and 2), with 10 explants per Petri dish. After pre-cultivation in the dark, the explants were cultured under light conditions, where the light intensity was 65 µmol m^−2^ s^−1^ during the 16 h photoperiod and the temperature was 25 ± 1 °C. The duration of both experiments was 2 months. In total, both experiments consisted of 1188 variants in triplicate (10 explants in each replicate), and a total of 35,640 explants were involved. During the experiment, the regeneration efficiency (E) was calculated as the quotient of the number of explants with developed shoots (Ne) divided by the total number of explants (No); the results are expressed as a percentage: E = (Ne/No)·100 [53].

To study the influence of various factors on the efficiency of regeneration (the culture media and the duration of the dark period of the explant pre-cultivation and shoot elongation) to increase the efficiency of regeneration, we used two grapevine genotypes (the Kober 5BB rootstock and the Podarok Magaracha cultivar), with those types of explants cultivated with the supplemental growth regulators (fragments of petioles cultivated on 2.0 mg L^−1^ BA and 0.15 mg L^−1^ IBA for the rootstock Kober 5BB and fragments of leaves cultivated on 2.0 mg L^−1^ BA and 0.05 mg L^−1^ for the cultivar Podarok Magaracha) that showed the highest efficiency of shoot formation in experiments no. 1 and 2. The leaves were cut into fragments measuring 5 mm × 5 mm and placed in Petri dishes on the adaxial side. Petioles and internodes were cut into fragments 5 mm long and placed longitudinally on the culture medium.

To study the influence of the culture media on the regeneration efficiency of the Kober 5BB rootstock and the Podarok Magaracha cultivar, six types of media were used: IM, PIV [20], NN [54], WPM [55], DKW [56], and MS. Explants were placed in Petri dishes containing culture medium (IM, PIV, NN, WPM, DKW, and MS, all with 3% (*w*/*v*) sucrose and 0.75% (*w*/*v*) agar) supplemented with 2.0 mg L^−1^ BA and 0.15 mg L^−1^ IBA for the rootstock Kober 5BB, and 2.0 mg L^−1^ BA and 0.05 mg L^−1^ for the cultivar Podarok Magaracha. All explants were pre-cultivated for 2 weeks under dark conditions. They were then transferred for cultivation to light conditions with a 16 h photoperiod (65 µmol m^−2^ s^−1^) at 25 ± 1 °C. The experiment was performed in duplicate, with 15 explants in each and a total of 360 explants used. The duration of the experiment was 8 weeks. During the experiment, the regeneration efficiency (E) was calculated as the quotient of the number of explants with developed shoots (Ne) divided by the total number of explants (No); the results were expressed as a percentage: E = (Ne/No)·100.

To study the effect of the duration of the dark phase of cultivation on the efficiency of regeneration, explants were placed in contact with IM with 3% (*w*/*v*) sucrose, 0.75% (*w*/*v*) agar, with the addition of the growth regulators 2.0 mg L^−1^ BA and 0.15 mg L^−1^ IBA for the rootstock Kober 5BB or 2.0 mg L^−1^ BA and 0.05 mg L^−1^ for the cultivar Podarok Magaracha. Explants were pre-cultured under dark conditions for 0, 1, 2, 3, 4, or 5 weeks. They were then transferred for cultivation to light conditions, with a 16 h photoperiod (65 µmol m^−2^ s^−1^) at 25 ± 1 °C. The experiment was performed in duplicate, with 15 explants in each and a total of 360 explants used. The duration of the experiment was 7 weeks. During the experiment, the regeneration efficiency and duration period of regeneration were observed. The regeneration efficiency (E) was calculated as the quotient of the number of explants with developed shoots (Ne) divided by the total number of explants (No); the results were expressed as a percentage: E = (Ne/No)·100. The duration of the regeneration period was considered to be the number of days between the appearance of the first and last regenerated plants.

### 4.3. Development of an Elongation Protocol

To study the effect of the duration of the dark phase in explant cultivation on the elongation efficiency, explants were pre-cultured under dark conditions for 2 weeks in IM with 3% (*w*/*v*) sucrose, 0.75% (*w*/*v*) agar, with the addition of the growth regulators, i.e., 2.0 mg L^−1^ BA and 0.15 mg L^−1^ IBA for the rootstock Kober 5BB and 2.0 mg L^−1^ BA and 0.05 mg L^−1^ IBA for the cultivar Podarok Magaracha. They were then transferred for cultivation to light conditions in the same media. Explants were cultivated in the light for 3 or 4 weeks. Then, they were transferred in the same media to dark conditions for the elongation of regenerated shoots. The experiment was performed in duplicate, with 15 explants in each and a total of 360 explants used. The total duration of the experiment was 14 weeks. During the experiment, the elongation efficiency (E) was calculated as the quotient of the number of explants with elongated (more than 1 cm in length) shoots (Ne) divided by the total number of explants with shoots (No); the results were expressed as a percentage: E = (Ne/No)·100.

Elongated plants and regenerated plants were cut from explants, grown, and rooted on PG medium with the addition of 0.05 mg L^−1^ NAA.

### 4.4. Agrobacterium-Mediated Transformation

The *A. tumefaciens* strain EHA105 [57] harboring the *pBin35SGFP* binary vector was used for the transformation experiments. The T-DNA of the *pBin35SGFP* binary vector contained the m-*gfp5*-ER and *npt*II genetic cassettes (Figure 11). Transformation experiments were carried out using pre-cultured leaf explants of the Podarok Magaracha grapevine cultivar.

#### 4.4.1. Transient Transformation

Experiments were conducted via the co-cultivation of leaf explants (pre-cultured leaf fragments for three days on IM, with the addition of the growth regulators (2.0 mg L^−1^ and 0.05 mg L^−1^ IBA)) jointly with *Agrobacterium tumefaciens* suspension. An overnight culture of *A. tumefaciens* was prepared in 50 mL of liquid yeast extract-bacto (YEB) medium (5 g L^−1^ beef extract, 5 g L^−1^ peptone, 5 g L^−1^ sucrose, 1 g L^−1^ yeast extract, 0.2 g L^−1^ MgSO4, pH 7.5 [58]), supplemented with 100 mg L^−1^ Km and 50 mg L^−1^ rifampicin (Rif), by shaking in an orbital shaker (180 rpm) for 24 h at 28 °C in the dark. The leaf explants were inoculated with five agrobacterial suspensions, diluted with distilled water to 0.2, 0.4, 0.6, 0.8, and 1.0 unit (OD_600_) per 300 explants for each concentration of inoculum. The explants were placed into glasses containing 50 mL of inoculum and kept for 30 min in an orbital shaker (100 rpm) for inoculation. After drying in an air-flow laminar box, the explants were transferred onto filter paper, placed on a solid surface of IM supplemented with 2.0 mg L^−1^ BA and 0.05 mg L^−1^ IBA in Petri dishes, and co-cultivated under dark conditions at a temperature of 24 °C for 24 h, 48, 72, 96, or 120 h. After the co-cultivation period, the explants were washed with distilled water supplemented with 300 mg L^−1^ Tm and then transferred to the medium for explant cultivation and the elimination of *Agrobacterium* (IM containing 2.0 mg L^−1^ BA, 0.05 mg L^−1^ IBA, 150 mg L^−1^ Tm, and 50 mg L^−1^ Km). Every e 2 weeks, the explants were transferred to fresh media. A total of 1500 leaf explants were used in the experiment (60 explants for each of 5 inoculum concentrations in 5 time periods of co-cultivation).

Transient *gfp* expression was detected using a SteREO Discovery V12-1 (Zeiss, GmbH, Jena, Germany), with the optical filter GFP Plus (GFP excitation at 450–490 nm). Real-time image capture was performed with an Axiocam 506 color (Zeiss, Germany) camera attached to the fluorescent microscope.

The number of explants expressing GFP was counted on 1, 2, 3, 4, 5, 6, 14, 21, 28, 42, 56, and 70 days after transformation. The transformation efficiency (E) was calculated as the quotient of the number of transient expression explants (Ne) divided by the total number of explants (No); the results were expressed as a percentage: E = (Ne/No)·100 [53].

#### 4.4.2. Stable Transformation

To optimize the combination of growth regulators for the regeneration of transgenic shoots, we used IM, supplemented with the growth regulator BA (ranging from 1.5 to 3.0 mg L^−1^ in increments of 0.5 mg L^−1^) jointly with IBA or IAA (each 0.1–0.4 mg L^−1^ with a step of 0.1 mg L^−1^). Thus, thirty-two combinations of growth regulators were used in the experiment, with 100 explants in each combination. To carry out transformations, the explants were pre-cultured for three days on IM supplemented with each of the 40 combinations of growth regulators, and then the explants were inoculated and co-cultured on the same media. Genetic transformation was carried out with an inoculum of the agrobacterial strain EHA105 with concentration at an OD_600_ = 0.8. Explants were co-cultivated under dark conditions at a temperature of 24 °C for 72 h. Further, the explants were washed with distilled water supplemented with 600 mg L^−1^ Tm (the first time) and 300 mg L^−1^ Tm (a second time) and then transferred to the same medium, for shoot regeneration, containing growth regulators and antibiotics for the elimination of *A. tumefaciens* (150 mg L^−1^ Tm) and the selection of transgenic tissue (50 mg L^−1^ Km). Every 2 weeks, the explants were transferred to fresh media. In sum, 4000 explants were inoculated.

### 4.5. PCR Analysis

For PCR analysis, the genomic DNA was isolated from antibiotic-resistant and wild-type grapevine plants using the cetyltrimethylammonium bromide (CTAB) method [59] Npt-mf-up, 5′-TCTGATGCCGCCGTGTTCC-3′ and Npt-mf-low, 5′-ATGCGCGCCTTGAGCCTG-3′ (anticipated amplification fragment of 448 bp); those for *gfp* were GFP-F, 5′-GGACGACGGGAACTACAAGA-3′ and GFP-R, 5′-CATGCCATGTGTAATCCCAG-3′ (anticipated amplification fragment of 350 bp); those for *virB* were virBf, 5′-GGCTACATCGAAGATCGTATGAATG-3′ and virBr, 5′-GACTATAGCGATGGTTACGATGTTGAC-3′ (anticipated amplification fragment of 480 bp). The PCRs for the *npt* ll, *virB*, and *gfp* genes were carried out in a 10 µL volume containing 1.0 µL of 10× Taq Turbo buffer, 0.2 µL of 10 mM deoxynucleotide triphosphates (dNTPs), 0.1 µL of two 100 ρM primers, 0.2 µL of 5 U µL^−1^ HS-Taq polymerase, 7.5 µL of mQ H_2_O, and 1 µL (≈30 ng) of a DNA template. Reactions were carried out in a Mastercycler nexus gradient (Eppendorf, Hamburg, Germany) as follows: for Npt-mf-up–Npt-mf-low and GFP-F–GFP-R, 1 cycle of 5 min at 95 °C, followed by 35 cycles of 15 sec at 95 °C, 30 sec at 60 °C, and 1 min at 72 °C, and 1 final cycle of 7 min at 72 °C, and, for virBf–virBr, 1 cycle of 5 min at 95 °C, followed by 35 cycles of 45 s at 93 °C, 45 s at 58 °C, and 45 s at 72 °C, and 1 final cycle of 5 min at 72 °C. The products were separated on 2.0% agarose gel using electrophoresis.

### 4.6. Southern Blot Analysis

Genomic DNA (10 μg) from grapevine transformed with *pBin35SGFP* was digested overnight at 37 °C with 100 U HindIII, which cuts out plant DNA. DNA from wild-type grapevine plants digested with HindIII served as a negative control. After agarose gel (0.8%) electrophoresis, the digestion products were transferred to, and immobilized on, a Hybond N+ membrane (Amersham Bioscience, Piscataway, NJ, USA) according to the manufacturer’s instructions. The DNA probes were synthesized via PCR using the plasmid *pBin35SGFP* as a template and the primers GFPup (5′-ATGGTGAGCAAGGGCGAGGAG-3′) and GFPlow (5′-TTACTTGTACAGCTCGTCCAT-3′). DNA probes (*gfp*—700 bp) were labeled with alkaline phosphatase using an AlkPhos Direct Labeling Kit (Amersham Bioscience, Piscataway, NJ, USA). The prehybridization, hybridization (overnight at 60 °C) with an alkaline phosphatase-labeled probe, and subsequent washings of the membrane were carried out according to the AlkPhos Direct Labeling Kit protocol. Detection was performed using CDP-Star detection reagent following the manufacturer’s instructions (Amersham Bioscience, Piscataway, NJ, USA).

### 4.7. Statistical Analysis

Significant differences between the variants were estimated by analysis of variance (ANOVA) followed by multiple comparisons of individual averages and evaluation by Duncan’s test using Statistica 6.1 (Dell Inc., Round Rock, TX, USA). All data presented as percentages were analyzed statistically after arcsine transformation. The data were also analyzed using the logistic regression method. Logistic regression models were built using the Logit class of the statsmodels library for python. The maximum likelihood estimation (MLE) method was used to train the models.

## 5. Conclusions

As a result of our research, we developed a direct organogenesis protocol for the cultivar Podarok Magaracha and the rootstock Kober 5BB. Using this protocol, the conditions for the agro-transformation of the Podarok Magaracha cultivar were developed, and two plants that showed the integration of the *pBin35SGFP* construct into their genome were obtained. Southern blot analysis revealed the presence of one-to-two transgene insertions in these GFP-expressing lines, which is typical in *Agrobacterium*-mediated transformation. The developed transformation procedure should enable the use of this grapevine cultivar for research on bioengineering and functional genomics in grapevines.

## Figures and Tables

**Figure 1 plants-13-02779-f001:**
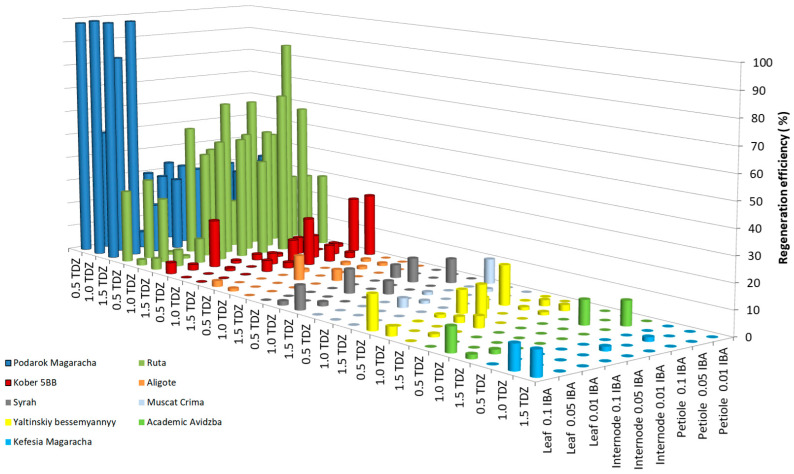
Regeneration efficiency of grapevine genotypes on MS medium supplemented with 2% (*w*/*v*) sucrose and a combination of TDZ and IBA.

**Figure 2 plants-13-02779-f002:**
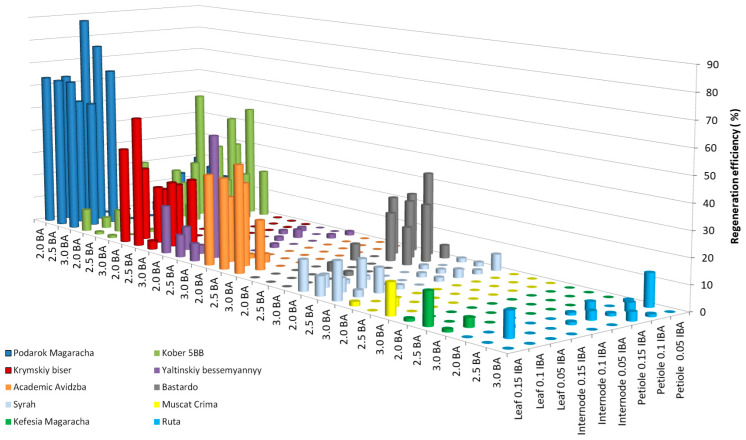
Regeneration efficiency of grapevine genotypes on IM medium supplemented with 3% (*w*/*v*) sucrose and a combination of BA and IBA.

**Figure 3 plants-13-02779-f003:**
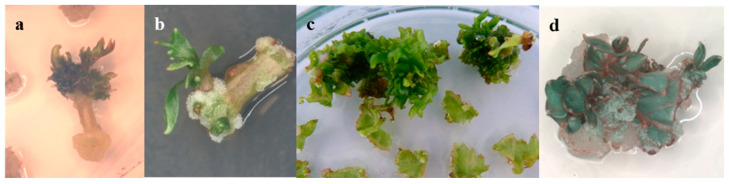
Effect of growth regulators and base medium on regenerants. Shoots derived from explants cultured on MS medium with added TDZ and IBA ((**a**)—from a petiole fragment explant of the Kober 5BB rootstock; (**c**)—from a leaf fragment explant of the Podarok Magaracha cultivar) often exhibited shoot deformation and vitrification, while those on IM medium supplemented with BA and IBA ((**b**)—from a petiole fragment explant of the Kober 5BB rootstock; (**d**)—from a leaf fragment explant of the Podarok Magaracha cultivar) displayed normal shoot formation.

**Figure 4 plants-13-02779-f004:**
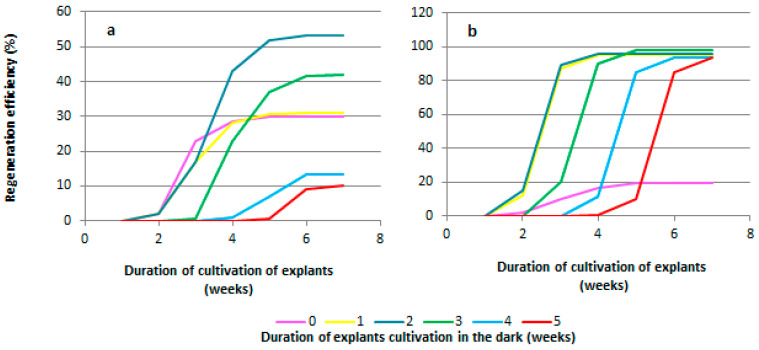
Effect of pre-cultivation under dark conditions on regeneration of most responsible genotypes. The duration of pre-cultivation in dark conditions significantly affects the efficiency of grapevine regeneration. This is evident in the dynamics of regeneration over seven weeks of cultivation for the Kober 5BB rootstock (**a**) and the Podarok Magaracha cultivar (**b**).

**Figure 5 plants-13-02779-f005:**
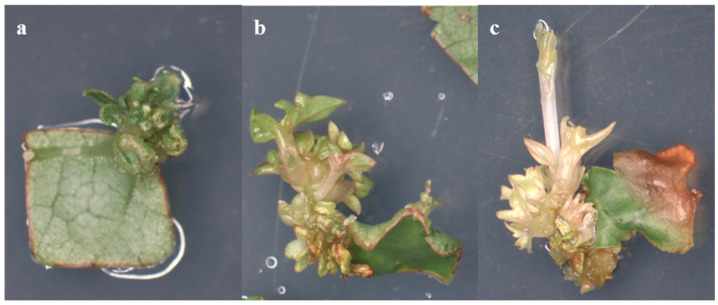
The influence of the duration of dark cultivation on the elongation of regenerated shoots of the Podarok Magaracha cultivar ((**a**)—0 days cultivation in the dark, (**b**)—21 days cultivation in the dark, (**c**)—35 days cultivation in the dark).

**Figure 6 plants-13-02779-f006:**
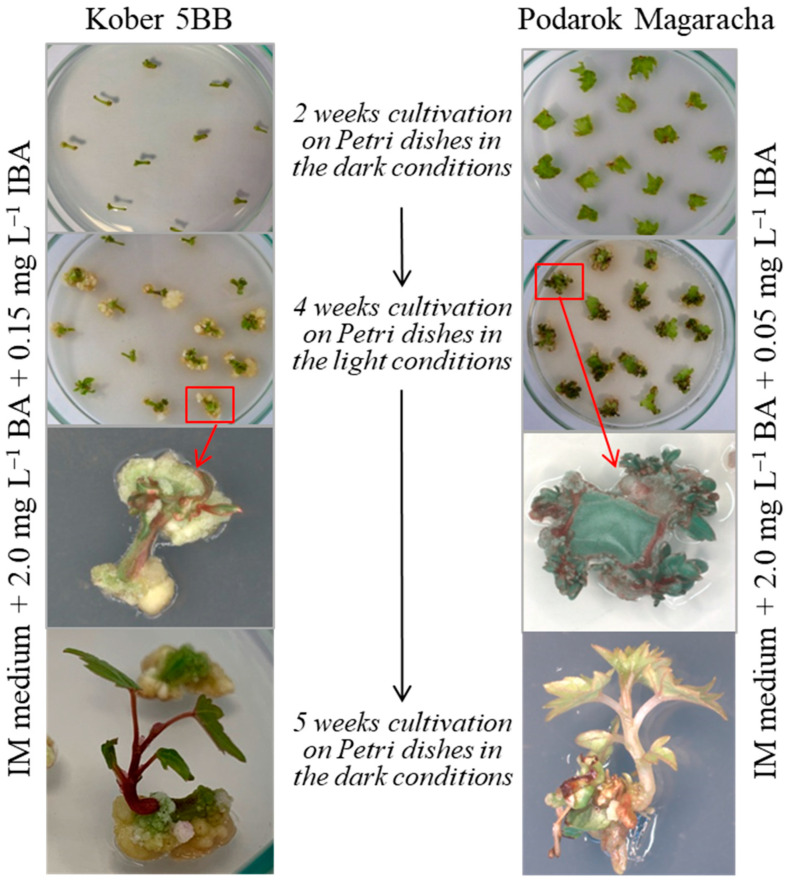
Tissue-culture cycle (direct organogenesis) for the two most responsible grapevine genotypes.

**Figure 7 plants-13-02779-f007:**
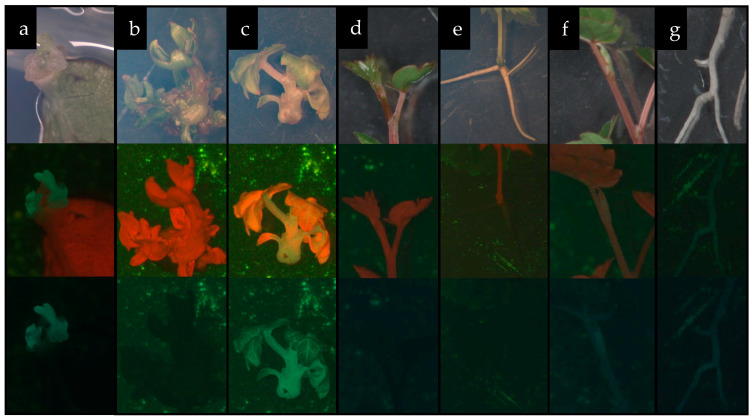
Development of control and genetically modified ‘Podarok Magaracha’ plants [(**a**)—meristematic buds expressing GFP; (**b**,**c**)—shoot regeneration in control (**b**) and GFP-expressing modified (**c**) plants; (**d**–**g**)—formation of complete control plants ((**d**)—shoot, (**e**)—root) and GFP-expressing modified plants ((**f**)—shoot, (**g**)—root)].

**Figure 8 plants-13-02779-f008:**
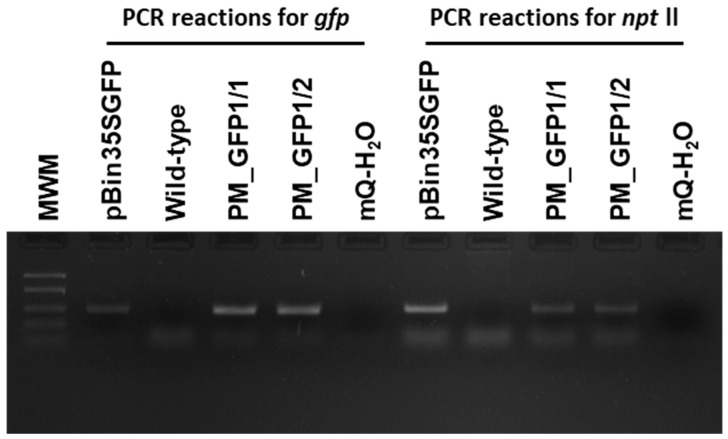
Primary molecular characterization of the generated ‘Podarok Magaracha’ transgenic lines. Polymerase chain reaction analyses for gfp and nptll transgenes in two independent Km-resistant regenerated individuals (PM_GFP1/1 and 1/2). Wild-type, non-transformed ‘Podarok Magaracha’ regenerated plant. MWM, molecular weight marker FastRulerTM Low Range DNA ladder (1500, 850, 400, 200, 50 bp).

**Figure 9 plants-13-02779-f009:**
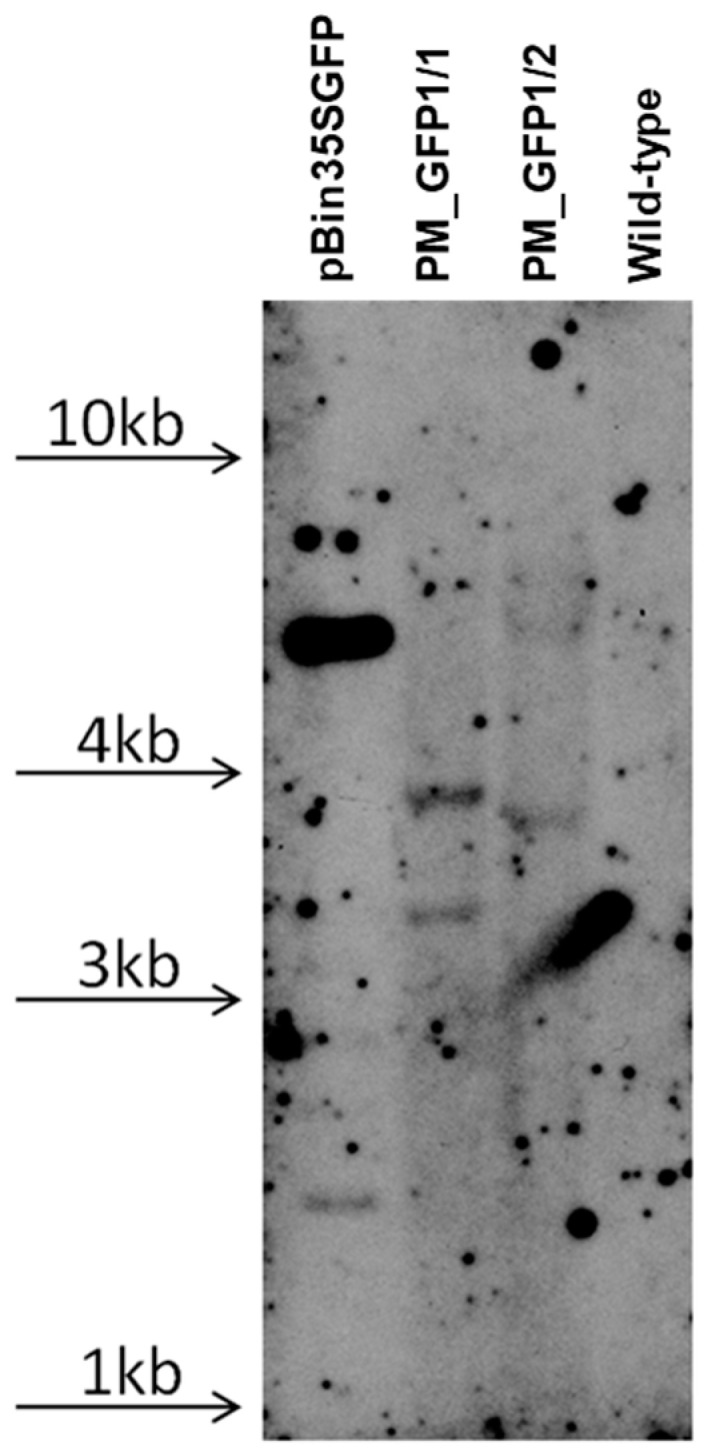
Southern blot analysis of transgenic grapevine lines. The hybridization result of probe to *gfp* gene followed by digesting the DNA with Hind III; the arrows indicate the location of the DNA on the gel of the appropriate size.

**Figure 10 plants-13-02779-f010:**
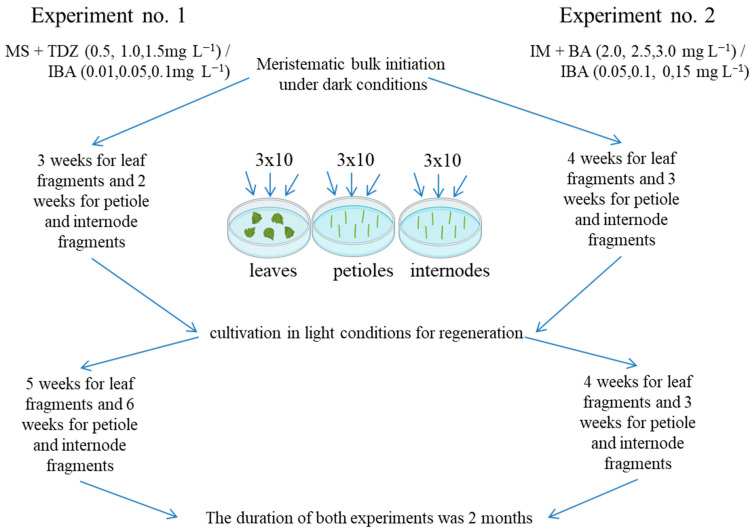
Scheme of the first two experiments on regeneration of 22 genotypes of grapevine.

**Figure 11 plants-13-02779-f011:**
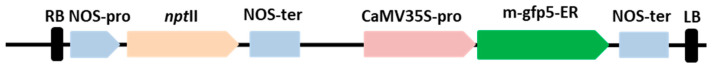
The vector construct includes a reporter codon-optimized *gfp* gene with an endoplasmic reticulum localization signal (m-*gfp5*-ER). The m-*gfp5*-ER gene is placed under the control of the CaMV35-p promoter and the NOSpolyA terminator. The selective gene for the synthesis of *npt*II is under the control of the regulatory elements of the nopaline synthase gene (NOS-pro and NOS-ter).

**Table 1 plants-13-02779-t001:** Influence of different basal culture media on the regeneration efficiency of the most responsible genotypes.

Genotype	MS	PIV	NN	DKW	WPM	IM
Kober 5BB	0 a	0 a	9.7 bc	18.0 c	1.7 ab	52.1 d
Podarok Magaracha	50.0 a	43.3 a	32.7 a	60.2 a	64.1 a	97.6 b

Regeneration efficiency data are presented in %. Different letters in a column indicate significant differences in variant data.

**Table 2 plants-13-02779-t002:** Transformation efficiency and influence of the duration of co-cultivation and *Agrobacterium* inoculum concentration in ‘Podarok Magaracha’ explants.

Duration of Cocultivation of Explants with *Agrobacterium* (h)	Inoculum Concentrations (OD_600_)	Duration of Cultivation of Explants on a Selective Medium (Days)												
		1	2	3	4	5	6	14	21	28	42	56	70	
24	0.2	0.0	15.5	17.2	24.1	24.1	22.4	17.2	10.3	8.6	8.6	6.9	6.9	bc
	0.4	0.0	19.0	27.6	41.4	44.8	32.8	25.9	20.7	7.0	3.5	3.5	3.5	a
	0.6	0.0	15.5	31.0	57.6	59.2	57.9	50.0	36.2	15.5	12.1	12.1	12.1	fghij
	0.8	0.0	39.7	46.6	60.3	65.5	44.8	27.6	19.0	10.3	8.6	6.9	6.9	bc
	1.0	0.0	25.9	43.1	55.2	53.5	39.7	27.6	15.5	10.3	10.3	6.9	6.9	bc
48	0.2	-	1.7	36.2	75.9	67.2	58.6	41.4	39.7	31.0	24.1	8.5	8.5	cdef
	0.4	-	10.3	58.6	82.8	79.3	74.1	48.3	34.5	19.0	10.5	8.5	8.5	cdef
	0.6	-	17.2	69.0	86.2	82.8	74.1	48.3	36.2	19.0	12.1	8.6	5.1	ab
	0.8	-	36.2	87.9	93.1	91.4	82.8	70.7	56.9	31.0	17.2	10.3	8.5	cdef
	1.0	-	19.0	46.6	67.2	60.3	48.3	37.9	32.8	22.4	13.5	10.3	10.3	efghi
72	0.2	-	-	32.8	74.1	67.2	50.0	36.6	49.1	22.4	19.0	17.2	17.2	klmn
	0.4	-	-	46.6	82.6	79.3	62.1	41.4	41.0	22.4	17.2	13.8	13.8	hijkl
	0.6	-	-	51.7	94.8	86.2	72.4	65.5	50.1	34.5	24.1	20.7	19.0	mn
	0.8	-	-	81.0	96.6	94.8	87.9	82.8	56.7	37.9	27.6	25.7	24.3	pq
	1.0	-	-	70.7	86.2	84.5	75.9	67.2	56.7	43.1	36.2	29.3	27.6	qr
96	0.2	-	-	-	68.4	79.3	77.6	56.7	34.5	25.9	22.4	19.0	17.2	mn
	0.4	-	-	-	82.8	82.8	77.6	65.5	46.6	37.9	19.0	17.2	13.8	ghijkl
	0.6	-	-	-	87.9	89.3	80.7	68.4	39.2	35.7	20.7	17.2	15.5	jklmn
	0.8	-	-	-	89.7	94.8	94.8	74.1	37.9	37.9	31.0	25.5	24.1	opq
	1.0	-	-	-	64.0	67.2	60.3	51.7	46.6	41.4	37.9	29.3	29.3	r
120	0.2	-	-	-	-	69.0	91.4	62.1	44.8	24.1	19.0	12.1	10.3	defghi
	0.4	-	-	-	-	84.1	91.4	65.5	41.4	19.0	13.8	8.5	8.5	cdef
	0.6	-	-	-	-	91.4	93.1	79.3	58.6	33.1	25.9	19.0	19.0	n
	0.8	-	-	-	-	94.8	93.1	75.9	50.0	39.1	31.0	10.3	10.3	fghi
	1.0	-	-	-	-	50.0	51.7	43.7	41.4	34.5	31.0	17.2	13.8	ijkl

The maximum transient transformation efficiency is indicated in red, and the transformation efficiency data are expressed as a percentage. In total, the experiment consisted of 25 variants in triplicate (20 explants in each replicate), and a total of 1500 explants were involved. Different letters indicate a reliable belonging of the data to a single (in the case of one letter) or a multiple (in the case of several letters) cluster, according to Duncan’s multiple range test. The best data for each experimental variant are highlighted in red.

**Table 3 plants-13-02779-t003:** Influence of various concentrations of indole-acetic acid (IAA), IBA and BA, and their combinations on the transformation efficiency of ‘Podarok Magaracha’.

BA Concentration, mg L^−1^	Number of GFP-Expressing Regenerated Plants (pcs per 100 Explants)									
	IAA Concentration (mg L^−1^)					IBA Concentration (mg L^−1^)				
	0.1	0.2	0.3	0.4	0.5	0.1	0.2	0.3	0.4	0.5
1.5	0 a	0 a	0 a	1 b	0 a	0 a	0 a	0 a	0 a	0 a
2.0	0 a	0 a	0 a	1 b	0 a	0 a	0 a	0 a	0 a	0 a
2.5	0 a	0 a	1 b	4 c	0 a	0 a	0 a	0 a	0 a	0 a
3.0	0 a	0 a	0 a	2 b	0 a	0 a	0 a	0 a	0 a	0 a

In total, the experiment consisted of 40 variants in quadruple repetition (25 explants in each replicate), and a total of 4000 explants were involved. Different letters in a column indicate significant differences in variant data, according to the Duncan’s multiple range tests.

## Data Availability

The data that support the tables and figures in this study are available from the corresponding author, upon reasonable request.

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
