# Peer review of "Organogenesis in a Broad Spectrum of Grape Genotypes and Agrobacterium-Mediated Transformation of the Podarok Magaracha Grapevine Cultivar"

_plants, 2024, doi:10.3390/plants13192779_

Round 1

Reviewer 1 Report

Comments and Suggestions for Authors

Dear authors,

First of all, I would like to congratulate you all for the extensive work you have done testing so many grapevine varieties, culture media, explant types, growth regulators, etc. Working with so many explants is hard and tiresome, but is the best way to obtain reproducible and robust results, which is, unfortunately, difficult to find in many published manuscripts.

Nonetheless, the manuscript presents several problems. First of all, I would recommend you to send the manuscript to an English editing service, because there are numerous errors throughout the whole manuscript, and at some points it is difficult to understand or follow the idea that is being described.

Introduction: which is the situation of GMOs in Russia? I think this is a critical point that should be discussed in the introduction

Results: I think most of this section should be rewritten. Basically, you have focused most of the results section on the effect of the grapevine variety, but you are not explaining the effect of other variables such as the explant type, hormone concentrations....etc.

Lines 73-82: the values given are for specific TDZ and IBA concentrations, not general values for each variety. Modify.

Tables 1 and 2: you are using different font for numbers and letters. Please, unify. These big tables with so many data are difficult to read and compare values among different treatments. I would really appreciate if you consider another way of presenting these results in a more visual and easy way.

Lines 83-87 and 105-109 are repeated.

Lines 144-151: is this problem caused by TDZ, or the problem is the use of MS basal medium or lower sucrose concentrations? You cannot state that the reason for vitrification is TDZ when you are not using the same basal medium and sucrose concentrations in both experiments. Modify

Lines 168-185 and so on: if the values are presented in the figures, avoid adding them in the text. It is repetitive information. Just compare and explain which conditions gave the best and worst results, and cite the corresponding table or figure if readers want to check the values. In case you want to give the values in the text, at least erase the b and d images in Figure 3.

The same for lines 194-210.

Lines 224-232: why do you need a callogenesis step if you are saying this is a direct organogenesis process??? Can you explain this?

Line 277: the number of the figure is incorrect

Discussion: overall, this section is too short and quite poor. It should be rewritten. It is more like a results section, rather than a discussion section.

Lines 359-366: this was not explained in the results section

Lines 368-375, 385-386, and so on: not a discussion, very poor. Please, perform an extensive bibliographic revision and compare your results with similar studies performed by other authors.

Mat&Met: one of the biggest problems in this section is that you have just presented regeneration % as the main trait for evaluating the success of the different treatments applied. For this manuscript to be suitable for publication, you should add extra information, such as the number of shoots produced per explant, the viability and length of those shoots, and it would also be of great interest to know what happens next, that's it, if those shoots originating from different treatments have the same rooting rates, survival, etc.

I also have a big concern regarding the statistical analysis. Most of the results presented are percentages. How did you perform ANOVA with those percentages? You don't have replicates if using percentages, right? That's why you are not presenting standard errors in Tables 1 and 2?. Or are you using a binomial (YES/NO) when analysing the regeneration results? In that case you should not perform ANOVA, but a logistic regression. Please, clarify this point.

Line 413: specify which are those grapevine varieties. I think it is more appropriate to use the term "variety" or "cultivar" rather than genotype. Within each variety you can find different genotypes.

Lines 456-457: please, reformulate, it is not clear.

Lines 462-464: which are those 2 varieties?

Which parameters were measured in section 4.2 to study the effect of all those media???

The experimental design here is not the best one: if you want to compare the effect of two different hormones, you should keep the same basal medium and sucrose concentrations. If not, the results obtained may be misleading. Please, be careful in the results and discussion sections when describing these results.

Lines 472-473: no sucrose? no agar?

Why not including PG medium in the induction and regeneration experiments???

Please, modify the title of sections 4.2 and 4.3 and create a new section for the dark experiments.

Line 497: IBA is missing

Lines 532-535: rewrite, difficult to understand.

Please, check abbreviations throughout the text, and add the full word the first time an abbreviation is written.

Comments on the Quality of English Language

Extensive editing of English language required

Author Response

Comments 1: Nonetheless, the manuscript presents several problems. First of all, I would recommend you to send the manuscript to an English editing service, because there are numerous errors throughout the whole manuscript, and at some points it is difficult to understand or follow the idea that is being described.

Response 1:If the reviewers approve the substantive changes made to the manuscript, they agree to send it to the English editing service.

Comments 2: Introduction: which is the situation of GMOs in Russia? I think this is a critical point that should be discussed in the introduction.

Response 2:In Russia there is currently a moratorium on growing GMOs in open ground. However, Russian legislation is not the subject of our research and does not affect the data presented in the article. I consider it inappropriate to describe this point in the introduction.

Comments 3: Results: I think most of this section should be rewritten. Basically, you have focused most of the results section on the effect of the grapevine variety, but you are not explaining the effect of other variables such as the explant type, hormone concentrations....etc.

Response 3:We agree with the comment. . Added to results section

Comments 4: Lines 73-82: the values given are for specific TDZ and IBA concentrations, not general values for each variety. Modify.

Response 4: We agree with the comment. . Added to results section

Comments 5: Tables 1 and 2: you are using different font for numbers and letters. Please, unify. These big tables with so many data are difficult to read and compare values among different treatments. I would really appreciate if you consider another way of presenting these results in a more visual and easy way.

Response 5:We agree with the comment. Tables are included in supplementary materials. The data is presented in two diagrams.

Comments 6: Lines 83-87 and 105-109 are repeated.

Response 6:Repeat removed

Comments 7: Lines 144-151: is this problem caused by TDZ, or the problem is the use of MS basal medium or lower sucrose concentrations? You cannot state that the reason for vitrification is TDZ when you are not using the same basal medium and sucrose concentrations in both experiments. Modify.

Response 7:We agree with the comment. The text has been changed «…the regenerants obtained on the MS medium with the addition of TDZ + IBA mostly had deformation and vitrification of the shoot…»

Comments 8: Lines 168-185 and so on: if the values are presented in the figures, avoid adding them in the text. It is repetitive information. Just compare and explain which conditions gave the best and worst results, and cite the corresponding table or figure if readers want to check the values. In case you want to give the values in the text, at least erase the b and d images in Figure 3.

Response 8:We agree with the comment. The picture has been changed.

Comments 9: The same for lines 194-210.

Response 9:We agree with the comment. The picture has been changed.

Comments 10: Lines 224-232: why do you need a callogenesis step if you are saying this is a direct organogenesis process??? Can you explain this?

Response 10:We agree with the comment. Terminology has been clarified. «The direct organogenesis protocol consists of culturing explants for 2 weeks in dark conditions for obtain meristematic bulk...».

Comments 11: Line 277: the number of the figure is incorrect.

Response 11: We agree with the comment. Number changed.

Comments 12: Discussion: overall, this section is too short and quite poor. It should be rewritten. It is more like a results section, rather than a discussion section.

Response 12:We agree with the comment. Corrections have been made.

Comments 13: Lines 359-366: this was not explained in the results section.

Response 13:We agree with the comment. Added to results section

Comments 14: Lines 368-375, 385-386, and so on: not a discussion, very poor. Please, perform an extensive bibliographic revision and compare your results with similar studies performed by other authors.

Response 14:We agree with the comment. Corrections have been made.

Comments 15: Mat&Met: one of the biggest problems in this section is that you have just presented regeneration % as the main trait for evaluating the success of the different treatments applied. For this manuscript to be suitable for publication, you should add extra information, such as the number of shoots produced per explant, the viability and length of those shoots, and it would also be of great interest to know what happens next, that's it, if those shoots originating from different treatments have the same rooting rates, survival, etc.

Response 15:The number of shoots per regenerating explant was 3-5. There were no statistically significant differences between the experimental variants. There was no further development of shoots. This required additional experiments outlined in the manuscript. Therefore, the only reliable indicator in the first experiments is the percentage of regeneration.

Comments 16: I also have a big concern regarding the statistical analysis. Most of the results presented are percentages. How did you perform ANOVA with those percentages? You don't have replicates if using percentages, right? That's why you are not presenting standard errors in Tables 1 and 2?. Or are you using a binomial (YES/NO) when analysing the regeneration results? In that case you should not perform ANOVA, but a logistic regression. Please, clarify this point.

Response 16:All variants of all experiments performed in triplicate (10 explants in each replicates) or duplicate (15 explants in each replicates) which makes it possible to apply analysis of variance. All data presented as percentages have been analyzed statistically after arcsine transformation. The text has been added «All data presented as percentages have been analyzed statistically after arcsine transformation».

Comments 17: Line 413: specify which are those grapevine varieties. I think it is more appropriate to use the term "variety" or "cultivar" rather than genotype. Within each variety you can find different genotypes.

Response 17:We agree with the comment. The text has been added «…20 cultivars (Merlot, Pinot Noir, Pinot Gris, Ruta, Sphinx, Cabernet Sauvignon, Aligote, Syrah, Bastardo, Muscat Blanc, Chardonnay, Podarok Magaracha, Kefesiya Magaracha, Akademik Avidzba, Yaltinskiy bessemyannyy, Muscat Crima, Veles, Tsitronnyy Magaracha, Livia, Krymskiy biser), one rootstock (Kober 5BB) and one breeding form (Magarach no. TT2,)…»

Comments 18: Lines 456-457: please, reformulate, it is not clear.

Response 18:We agree with the comment. The text has been changed « In total, both experiments consisted of 1188 variants in triplicate (10 explants in each replicates), for a total were involved of 35,640 explants».

Comments 19: Lines 462-464: which are those 2 varieties?

Response 19:We agree with the comment. Varieties has been clarified.

Comments 20: Which parameters were measured in section 4.2 to study the effect of all those media???

Response 20:We agree with the comment. The text has been added « During the experiment regeneration efficiency (E) was calculated as the quotient of dividing the number of explants with developed shoots (Ne) by the total number of explants (No); the results were expressed as a percentage: E = (Ne/No) 100 [56] ».

Comments 21: The experimental design here is not the best one: if you want to compare the effect of two different hormones, you should keep the same basal medium and sucrose concentrations. If not, the results obtained may be misleading. Please, be careful in the results and discussion sections when describing these results.

Response 21:We agree with the comment. The text has been changed «…the regenerants obtained on the MS medium with the addition of TDZ + IBA mostly had deformation and vitrification of the shoot…»

Comments 22: Lines 472-473: no sucrose? no agar?

Response 22:We agree with the comment. The text has been added «... all with 3% (w/v) sucrose, 0.75% (w/v) agar…»

Comments 23: Why not including PG medium in the induction and regeneration experiments???

Response 23:Thank you for your comment. We will try this in future studies.

Comments 24: Please, modify the title of sections 4.2 and 4.3 and create a new section for the dark experiments.

Response 24:We agree with the comment. Section names have been changed.

Comments 25: Line 497: IBA is missing

Response 25:We agree with the comment. The text has been added.

Comments 26: Lines 532-535: rewrite, difficult to understand.

Response 26:We agree with the comment. The text has been changed.

Comments 27: Please, check abbreviations throughout the text, and add the full word the first time an abbreviation is written.

Response 27:We agree with the comment. Abbreviations checked.

Reviewer 2 Report

Comments and Suggestions for Authors

This manuscript reports on searching simpler methods and conditions for the in vitro organogenesis of grapevine using 22 genotypes and improved methods of Agrobacterium-mediated transformation of Podarok Magaracha grapevine. The data of each experiment are presented in the manuscript. The information will be useful in developing more suitable methods and techniques for grapevine in vitro culture and genetic transformation in the future. However, I have several questions and suggestions for the Tables and Figures in the manuscript.

1. For all Tables: All data in the tables are displayed by percentage (%).   However, ANOVA and Duncan’s test were applied to indicate the significant differences. The % data should be analyzed statistically after arcsine transformation. ANOVA and Duncan’s test are probably not applicable. Please reconsider the statistical analysis for all tables. I understood that all experiments used in implants per Petri dish with triplication (3 replications) for each treatment (or sample).

2. Figure 1:  I couldn’t understand the purpose of applying principal component analysis (PCA) in this study. What is indicated by PCA's grouping or scattering of grapevine cultivars? I suggest that Figure 1 and its related text will be removed.

3. Figure 3:  There are four panels (a~d) in Figure 3. The data shown from panels a/b and c/d are dual presentation. The line charts (panels a and c) seem strange and abstracted (modeled curve lines).  I suggest that panels a and c be removed.

 Minor errors

P.2 L. 65; scientific literature] toward  >> scientific literature toward    * remove “ ] ”

P.5 L.238; Agrobacterium hybridized with >> Agrobacterium harboring

P.12 L.509; harbouring  >>  harboring

P.14 L.585; Genomic DNA from grapevine (10 µg) transformed  >> Genomic DNA (10 µg)  from grapevine transformed 

Author Response

Comments 1:For all Tables: All data in the tables are displayed by percentage (%).   However, ANOVA and Duncan’s test were applied to indicate the significant differences. The % data should be analyzed statistically after arcsine transformation. ANOVA and Duncan’s test are probably not applicable. Please reconsider the statistical analysis for all tables. I understood that all experiments used in implants per Petri dish with triplication (3 replications) for each treatment (or sample).

Response 1: All variants of all experiments performed in triplicate (10 explants in each replicates) or duplicate (15 explants in each replicates) which makes it possible to apply analysis of variance. All data presented as percentages have been analyzed statistically after arcsine transformation. The text has been added «All data presented as percentages have been analyzed statistically after arcsine transformation».

Comments 2. Figure 1:  I couldn’t understand the purpose of applying principal component analysis (PCA) in this study. What is indicated by PCA's grouping or scattering of grapevine cultivars? I suggest that Figure 1 and its related text will be removed.

Response 2: We agree with the comment. PCA changed.

Comments 3. Figure 3:  There are four panels (a~d) in Figure 3. The data shown from panels a/b and c/d are dual presentation. The line charts (panels a and c) seem strange and abstracted (modeled curve lines).  I suggest that panels a and c be removed.

Response 3: We agree with the comment. The picture has been changed.

 Minor errors

Comments 4: P.2 L. 65; scientific literature] toward  >> scientific literature toward    * remove “ ] ”

Response 4: We agree with the comment. Symbol deleted.

Comments 5: P.5 L.238; Agrobacterium hybridized with >> Agrobacterium harboring.

Response 5: We agree with the comment. The text has been changed.

Comments 6: P.12 L.509; harbouring  >>  harboring.

Response 6: We agree with the comment. The text has been changed.

Comments 7: P.14 L.585; Genomic DNA from grapevine (10 µg) transformed  >> Genomic DNA (10 µg)  from grapevine transformed .

Response 7: We agree with the comment. The text has been changed.

Round 2

Reviewer 1 Report

Comments and Suggestions for Authors

Dear authors,

Thank you for all the changes performed along the manuscript. However, the manuscripts still presents some weaknesses that should be checked, especially in the Results and Discussion sections.

Lines 73-83: as I already mentioned in the previous review process, the values here presented are not average values for all the treatments within one cultivar. Please, provide average values, not specific values from just one treatment. 

Figures 1 and 2: Indicate which is the basal medium and the sucrose concentration used for each one. Which are the criteria used not to show some varieties in the figure? Explain in the text please. If possible, put the cultivars in the same order in both figures, and use the same colours for each cultivar in both figures. Please, modify the legend of the figures. It is not necessary to put the same colour repeated three times for each of the treatments (redundant information). 

Lines 84-158: please, revise; there are numerous mistakes along the paragraph, like for example: the combination of IM medium and TDZ does not exist. 

Lines -158: order ideas and rewrite; I think you should start, first, mentioning aal those cultivars that present very low regeneration values (maybe < 10-15%???) or/and that do not present/follow a clear pattern (regeneration % vs explant type vs hormone concentration). Then, focus on the remaining varieties and just mention which factors have a significant impact on regeneration %, and under which conditions the best results were obtained. Do not give long explanations for factors that do not have a significant impact on the success of the process. For example: for IM medium and Bastardo variety, the explant type was the most influencing factor. For Academic, both the explant type (leaves) and hormones (IBA) had a significant impact. Try to extract general conclusions, patterns, etc. Try to group varieties that follow similar patterns. 

Lines 168-173: are these general values? It is repetitive with Lines 73-83. Or are for specific treatments?

Ruta: explain its exceptional performance in MS + TDZ

Figure 4 a): the blue colour here is different from the blue colour in b). The line for each of the treatments should start at the point when explants were transferred to light. 

Line 251: this is not correct. All explants were not cultured 7 weeks under light conditions.

Lines 275-282: this paragraph should have its own tittle. 

Discussion: some parts still are too generic and simplistic. No discussion for the performance of varieties which presented interesting results, such as Podarock Magaracha, Krymskiy, Academic, Ruta, Bastardo. Maybe you could discuss the similar behaviour observed for Krymskiy and Academic. They are hybrids, have you checked if they have a common origin that could explain the results?

Lines 401-404: these varieties have not been studied in this work. Maybe you can search for better examples for the use of BA. 

Any explanation for the low regeneration values observed for many "famous" and well-studied varieties (Pinot noir, Chardonnay, Cabernet sauvignon...)???

Lines 574-583: reformulate so the different dark periods are clearer for the reader (5, 6 weeks?)

Based don Response 15, authors should include this information in the manuscript, even if not statistically significant differences were observed.

Response 16 (Lines 681-686): the statistical analysis should be repeated. It is not appropriate to perform ANOVA with just 2 replicates (3 is the bare minimum). Additionally, you should not perform ANOVA with percentages (at least you have very big n). In your case, you have 10 or 15 explants, so you can only obtain 10 or 15 percentage values (10%, 20%, 30%, 40%, 50%, 60%, 70%, 80%, 90% and 100%, when using 10 explants). You don’t have a continuous series of numbers. In this case you should analyse your data as a binomial (YES/NO) and perform a logistic regression. Please, repeat all the analyses and, if required, modify the results section.

Comments on the Quality of English Language

Authors should send the manuscript to an English editing service to make the review process easier.

Author Response

Comments 1: Lines 73-83: as I already mentioned in the previous review process, the values here presented are not average values for all the treatments within one cultivar. Please, provide average values, not specific values from just one treatment. 

Response 1: Text has been changed. The corresponding indicators are included in tables S1 and S2.

Comments 2: Figures 1 and 2: Indicate which is the basal medium and the sucrose concentration used for each one. Which are the criteria used not to show some varieties in the figure? Explain in the text please. If possible, put the cultivars in the same order in both figures, and use the same colours for each cultivar in both figures. Please, modify the legend of the figures. It is not necessary to put the same colour repeated three times for each of the treatments (redundant information). 

Response 2: We agree with the comment. Data on basal medium and sucrose have been added. Criteria for excluding certain varieties have been added to the text of the manuscript. Arranging the varieties in both pictures in the same order is not possible due to the overlap of the bars in the diagram. varieties are arranged in order of decreasing regenerative ability. The legend to the figures has been changed.

Comments 3: Lines 84-158: please, revise; there are numerous mistakes along the paragraph, like for example: the combination of IM medium and TDZ does not exist. 

Response 3: We agree with the comment. The text has been changed

Comments 4: Lines -158: order ideas and rewrite; I think you should start, first, mentioning aal those cultivars that present very low regeneration values (maybe < 10-15%???) or/and that do not present/follow a clear pattern (regeneration % vs explant type vs hormone concentration). Then, focus on the remaining varieties and just mention which factors have a significant impact on regeneration %, and under which conditions the best results were obtained. Do not give long explanations for factors that do not have a significant impact on the success of the process. For example: for IM medium and Bastardo variety, the explant type was the most influencing factor. For Academic, both the explant type (leaves) and hormones (IBA) had a significant impact. Try to extract general conclusions, patterns, etc. Try to group varieties that follow similar patterns. 

Response 4: We agree with the comment. The text has been added « In general, all varieties can be clustered according to the effectiveness of their regenerative ability. The first cluster consists of 14 genotypes (Livia, Veles, Tsitronnyy Magaracha, Muscat Blanc, Sphinx, Aligote, Pinot Noir,  Merlot, Chardonnay, Cabernet Sauvignon, Pinot Gris, Kefesiya Magaracha, Muscat Crima and Magarach no. TT2). The regeneration efficiency of these genotypes is at a low level (no more than 13.0%), regardless of the combination of growth regulators used. The second cluster consists of 5 cultivars (Syrah, Bastardo, Akademik Avidzba, Krymskiy biser and Yaltinskiy bessemyannyy). These cultivars regenerate at an middle level when using combinations BA + IBA (regeneration efficiency from 15.0 to 53.0%), and at a low level when using combinations of growth regulators TDZ + IBA (regeneration efficiency from 3.5 to 10.0%). Rootstock Kober 5BB has a high regenerative ability in the presence of growth regulators BA + IBA (53.3%), in the presence of TDZ + IBA, the regeneration efficiency decreases to 25.5% (see Table S1 and Table S2, data is valid for petiole fragments as explants). The Podarok Magaracha cultivar is characterized by high regenerative ability both using a combination of growth regulators BA + IBA and using TDZ + IBA, the regeneration efficiency is 87.4 and 100.0%, respectively (see Table 1 and Table 2, data is valid for leaf fragments as explants). The Ruta cultivar has a high regenerative ability when using a combination of growth regulators TDZ + IBA (87%), and low regenerative ability in the presence of BA + IBA (13.3%) (see Table S1 and Table S2, data is valid for petiole fragments as explants)»

Comments 5: Lines 168-173: are these general values? It is repetitive with Lines 73-83. Or are for specific treatments?

Response 5: Text has been changed. The corresponding indicators are included in tables S1 and S2.

Comments 6: Ruta: explain its exceptional performance in MS + TDZ

Response 6: Apparently this is due to varietal characteristics. Since no special studies have been carried out, it is not possible to reliably explain this effect.

Comments 7: Figure 4 a): the blue colour here is different from the blue colour in b). The line for each of the treatments should start at the point when explants were transferred to light. 

Response 7: a) the color has been corrected. b) the line for each treatment starts from the beginning of the experiment, since in some cases regeneration could occur even when cultivated in the dark.

Comments 8: Line 251: this is not correct. All explants were not cultured 7 weeks under light conditions.

Response 8: We agree with the comment. The text has been changed

Comments 9: Lines 275-282: this paragraph should have its own tittle. 

Response 9: We agree with the comment. The text has been changed

Comments 10: Discussion: some parts still are too generic and simplistic. No discussion for the performance of varieties which presented interesting results, such as Podarock Magaracha, Krymskiy, Academic, Ruta, Bastardo. Maybe you could discuss the similar behaviour observed for Krymskiy and Academic. They are hybrids, have you checked if they have a common origin that could explain the results?

Response 10: All genotypes are unrelated. We did not analyze genomes in the work, so we have no justification for discussing similar reactions.

Comments 11: Lines 401-404: these varieties have not been studied in this work. Maybe you can search for better examples for the use of BA. 

Response 11: In order for the manuscript to have relevance and scientific novelty, we studied other varieties. The discussion section provides the opportunity to compare our results with those of other researchers, even using other genotypes cultivated under similar conditions. direct organogenesis is a minor method for grape regeneration. Therefore, the number of examples of using BA for grape regeneration is not so numerous. If you know of such examples, we would be grateful for the link provided.

Comments 12: Any explanation for the low regeneration values observed for many "famous" and well-studied varieties (Pinot noir, Chardonnay, Cabernet sauvignon...)???

Response 12: As a rule, the effectiveness of regeneration is associated with the properties of the genotype. we agree with this opinion. No specialized genome studies were carried out as part of the work. We have no basis for theorizing.

Comments 13: Lines 574-583: reformulate so the different dark periods are clearer for the reader (5, 6 weeks?)

Response 13: We agree with the comment. The text has been changed

Comments 14: Based don Response 15, authors should include this information in the manuscript, even if not statistically significant differences were observed.

Response 14: Since no statistically significant differences were found, we do not consider ourselves entitled to draw any conclusions. the text of the manuscript contains " There was no statistically significant effect of the duration of cultivation under lighting conditions (3 or 4 weeks of cultivation) on the efficiency of shoot elongation.»

Comments 15: Response 16 (Lines 681-686): the statistical analysis should be repeated. It is not appropriate to perform ANOVA with just 2 replicates (3 is the bare minimum). Additionally, you should not perform ANOVA with percentages (at least you have very big n). In your case, you have 10 or 15 explants, so you can only obtain 10 or 15 percentage values (10%, 20%, 30%, 40%, 50%, 60%, 70%, 80%, 90% and 100%, when using 10 explants). You don’t have a continuous series of numbers. In this case you should analyse your data as a binomial (YES/NO) and perform a logistic regression. Please, repeat all the analyses and, if required, modify the results section.

Response 15: In our research we were guided by generally accepted methodological recommendations De Groot MH (1970) Optimal statistical decisions. McGraw-Hill seriesin probability and statistics, New York, pp 45–347. According to the method for analysis of variance, at least 30 samples are required in repetitions (i.e. 3x10, 2x15, 5x6, etc.). Analyzing percentage values ​​after arcsine conversion is a standard and valid technique, as evidenced by a large number of publications, for example:

  • Xie, X.; Aguero, C. B.; Wang, Y.; Walker, A.M. Genetic transformation of grape varieties and rootstocks via organogenesis. 758 Plant Cell, Tissue Organ Cult 2016, 126, 541–552
  • Ehab, M.R.M.; Hemaid, I.A.S.; Omar, A.A.; Naif, M.K. Producing Transgenic Thompson Seedless' Grape (Vitis vinifera L.) 783 plants using Agrobacterium tumefaciens. International Journal of Agriculture and Biology 2016, 18, 661-670. 784 39.
  • Kumsa, F. Effect of growth regulators on indirect organogenesis of two grapevines (Vitis vinifera L.) cultivars. African Journal of 785 Biotechnology 2017, 16, 852-859.
  • Kumsa, F.; Feyissa, T. In vitro regeneration of two grapevine (Vitis vinifera L.) varieties from leaf explants. African Journal of 793 Biotechnology 2019, 18(4), 92-100.
  • Forleo, L.R.; D’Amico, M.; Basile, T.; Marsico, A.D.; Cardone, M.F.; Maggiolini, F.A.M.; Velasco, R.; Bergamini, C. Somatic Embryogenesis in Vitis for Genome Editing: Optimization of Protocols for Recalcitrant Genotypes. Horticulturae 2021, 7, 511. https://doi.org/3390/horticulturae7110511
  • Dhekney, S.A., Li, Z.T., Dutt, M. et al.Agrobacterium-mediated transformation of embryogenic cultures and plant regeneration in Vitis rotundifolia (muscadine grape). Plant Cell Rep 27, 865–872 (2008). https://doi.org/10.1007/s00299-008-0512-2

Reviewer 2 Report

Comments and Suggestions for Authors

All the points I commented on have been revised and improved. I couldn't see the comments from other reviewers. However, I believe the figures have been modified following their suggestions. 

Author Response

Thank you very much for your attention to our manuscript.

Round 3

Reviewer 1 Report

Comments and Suggestions for Authors

I have checked the manuscript, and authors have considerably improved the English. Nonetheless, authors have not repeated the statistical analysis as previously suggested. Authors have performed ANOVA with three or even two percentages (% obtained from groups of 10 or 15 explants) per condition. As I have already explained to the authors, performing ANOVA with so few replicates is not appropriate, as the power of the analysis decreases, together with the reliability of the results. Authors allege that they are following the Central Limit Theorem (CLT), which suggests that at least 30 samples are required in repetitions of, for example, 3 x 10 or 2 x 15. It is true that authors have used 30 samples (in fact, one of the strengths of this manuscript is the huge amount of explants and treatments that authors have used), but the problem is that they are not performing a statistical analysis with those 30 samples, they are extracting percentages in groups of 2 or 3 from those 30 samples, and they are statistically analysing those 2 or 3 percentages, which is not correct. The best procedure would be to analyse the data as a binomial (YES/NO) and perform a logistic regression. In that case they would be following the CLT.

Additionally, for the transformation experiments, they are using 60 and 100 explants per condition, but no information is provided about replicates and how those percentages are calculated and analysed.

As a result, I recommend sending the manuscript back to the authors to redo the statistical analysis, and if required, modify the results section accordingly. 

All the best,

Comments on the Quality of English Language

I have checked the manuscript, and authors have considerably improved the English. Nonetheless, authors have not repeated the statistical analysis as previously suggested. Authors have performed ANOVA with three or even two percentages (% obtained from groups of 10 or 15 explants) per condition. As I have already explained to the authors, performing ANOVA with so few replicates is not appropriate, as the power of the analysis decreases, together with the reliability of the results. Authors allege that they are following the Central Limit Theorem (CLT), which suggests that at least 30 samples are required in repetitions of, for example, 3 x 10 or 2 x 15. It is true that authors have used 30 samples (in fact, one of the strengths of this manuscript is the huge amount of explants and treatments that authors have used), but the problem is that they are not performing a statistical analysis with those 30 samples, they are extracting percentages in groups of 2 or 3 from those 30 samples, and they are statistically analysing those 2 or 3 percentages, which is not correct. The best procedure would be to analyse the data as a binomial (YES/NO) and perform a logistic regression. In that case they would be following the CLT.

Additionally, for the transformation experiments, they are using 60 and 100 explants per condition, but no information is provided about replicates and how those percentages are calculated and analysed.

As a result, I recommend sending the manuscript back to the authors to redo the statistical analysis, and if required, modify the results section accordingly. 

All the best,

Author Response

Comments 1: 

I have checked the manuscript, and authors have considerably improved the English. Nonetheless, authors have not repeated the statistical analysis as previously suggested. Authors have performed ANOVA with three or even two percentages (% obtained from groups of 10 or 15 explants) per condition. As I have already explained to the authors, performing ANOVA with so few replicates is not appropriate, as the power of the analysis decreases, together with the reliability of the results. Authors allege that they are following the Central Limit Theorem (CLT), which suggests that at least 30 samples are required in repetitions of, for example, 3 x 10 or 2 x 15. It is true that authors have used 30 samples (in fact, one of the strengths of this manuscript is the huge amount of explants and treatments that authors have used), but the problem is that they are not performing a statistical analysis with those 30 samples, they are extracting percentages in groups of 2 or 3 from those 30 samples, and they are statistically analysing those 2 or 3 percentages, which is not correct. The best procedure would be to analyse the data as a binomial (YES/NO) and perform a logistic regression. In that case they would be following the CLT.

Additionally, for the transformation experiments, they are using 60 and 100 explants per condition, but no information is provided about replicates and how those percentages are calculated and analysed.

As a result, I recommend sending the manuscript back to the authors to redo the statistical analysis, and if required, modify the results section accordingly. 

Responce 1: We conducted additional statistical analysis.